# Obtaining and maintaining cortical hand representation as evidenced from acquired and congenital handlessness

Daan B Wesselink[1,2†], Fiona MZ van den Heiligenberg[1,2†], Naveed Ejaz[3,4], Harriet Dempsey-Jones[1,2], Lucilla Cardinali[3,5], Aurelie Tarall-Jozwiak[6], Jörn Diedrichsen[3,4], Tamar R Makin[1,2,7*]

[1]Institute of Cognitive Neuroscience, University College London, London, United Kingdom; [2]Wellcome Centre for Integrative Neuroimaging, University of Oxford, Oxford, United Kingdom; [3]Brain and Mind Institute, University of Western Ontario, London, Canada; [4]Department of Computer Science, University of Western Ontario, London, Canada; [5]Unit for Visually Impaired People, Istituto Italiano di Tecnologia, Genoa, Italy; [6]Queen Mary's Hospital, London, United Kingdom; [7]Wellcome Centre for Human Neuroimaging, University College London, London, United Kingdom

**Abstract** A key question in neuroscience is how cortical organisation relates to experience. Previously we showed that amputees experiencing highly vivid phantom sensations maintain cortical representation of their missing hand (Kikkert et al., 2016). Here, we examined the role of sensory hand experience on persistent hand representation by studying individuals with acquired and congenital hand loss. We used representational similarity analysis in primary somatosensory and motor cortex during missing and intact hand movements. We found that key aspects of acquired amputees' missing hand representation persisted, despite varying vividness of phantom sensations. In contrast, missing hand representation of congenital one-handers, who do not experience phantom sensations, was significantly reduced. Across acquired amputees, individuals' reported motor control over their phantom hand positively correlated with the extent to which their somatosensory hand representation was normally organised. We conclude that once cortical organisation is formed, it is remarkably persistent, despite long-term attenuation of peripheral signals.
DOI: https://doi.org/10.7554/eLife.37227.001

*For correspondence:
t.makin@ucl.ac.uk

†These authors contributed equally to this work

## Introduction

A fundamental organising principle in the primary somatosensory cortex (SI) is somatotopic mapping, where adjacent body parts are represented more proximally on the cortical sheet than those further apart (*Penfield and Rasmussen, 1950*). In the cortical hand area, this topographic organising principle results in a detailed digit map (*Kaas et al., 1979*; *Penfield and Rasmussen, 1950*), where neighbouring digits on the hand are represented closer together on the neocortex than non-neighbouring digits, as can be shown with functional MRI (fMRI) (*Kolasinski et al., 2016*). More generally, cortical activity patterns for neighbouring fingers overlap more – and are therefore more similar – than non-neighbouring fingers, independent of the exact spatial arrangement of these patterns (*Ejaz et al., 2015*). The representational structure (i.e. the relative dissimilarity of activity patterns for different movements) is thought to reflect the natural statistics of hand use over one's life course (*Graziano and Aflalo, 2007*; *Overduin et al., 2012*): Concurrent inputs to neighbouring digits will increase representational similarity (*Wang et al., 1995*), while greater individuation of inputs will induce greater representational dissimilarity (*Ejaz et al., 2015*). Some have even suggested that

alterations of these inter-digit representational boundaries after the hand map has been formed, for example due to digit overuse, can disrupt (musician's dystonia [*Elbert et al., 1998*], but see *Ejaz et al., 2016*) or improve perception and action (tactile discrimination [*Recanzone et al., 1992*; *Pleger et al., 2003*]).

We (*Kikkert et al., 2016*) and others (*Flesher et al., 2016*; *Bruurmijn et al., 2017*) recently challenged the view that structured input from the periphery is required for preserving sensorimotor hand representation (*Dempsey-Jones et al., 2016*; see also *Davis et al., 1998*, *Mercier et al., 2006*, *Garbarini et al., 2018* for related brain stimulation and behavioural findings). We took advantage of a well-documented phenomenon whereby some amputees report being able to volitionally move their phantom hand, resulting in kinaesthetic phantom sensations (*Henderson and Smyth, 1948*). Phantom limb movements have been shown to elicit both central and peripheral motor signals that are different from those found during movement imagery (*Reilly et al., 2006*; *Raffin et al., 2012a*; *Raffin et al., 2012b*; *Makin et al., 2013b*). Using 7T imaging we explored whether three amputees experiencing exceptionally vivid phantom sensations maintained the canonical hand representation, exemplified by somatotopically organised representation of individual digits. We found that although digit selectivity was reduced, digit order and the extent of the missing hand maps in SI were similar to what is observed in controls.

Our previous findings demonstrate the stability of SI hand organisation despite decades of amputation. It remains unknown, however, whether hand representation after amputation reflects phantom sensations, and as such only persists in individuals with highly vivid phantom sensations. Here, we asked whether persistent representation of a missing hand reflects an organisational principle in the sensorimotor cortex, and thus will even be observed in amputees with little phantom sensations. To address this question, we measured cortical hand representation in 18 acquired amputees with varying vividness of their phantom sensations (hereafter amputees). To test the idea that the development of a hand representation requires sensory experience, we also tested 13 individuals missing one hand from birth (due to congenital amelia; hereafter congenital one-handers). All participants underwent fMRI while performing a visually cued motor task involving individual digit movements (of both the missing hand and the intact hand). Activity patterns in the missing hand of SI and M1 were analysed using representational similarity analysis (*Walther et al., 2016*; *Diedrichsen et al., 2016*). We hypothesised that normal peripheral input is necessary to establish normal sensory hand representation, but not to maintain it.

## Results

### Phantom hand movements elicit typical hand representation in the missing hand area of acquired amputees

We first focused our analysis on the representation of the missing hand, as revealed by instructing individuals to move individual digits of their missing hand (or nondominant hand in controls). We interrogated fMRI activity in the SI hand area contralateral to the missing/nondominant hand (see Materials and methods for regions of interest (ROI) definition). We examined univariate task-related activity, as quantified by averaging the BOLD response across all the digit conditions within the missing hand ROIs (*Figure 1A*, see *Figure 1—figure supplement 1* for M1 ROI results). Overall, all participants, including congenital one-handers, were able to engage the missing/nondominant hand area to some degree. Although activity was reduced in SI for the congenital one-handers' missing hand compared to controls (t(23)=3.5, p=0.002), activity was significantly greater than baseline (t(12)=2.6, p=0.02).

To investigate digit discriminability in the hand area, we next estimated the dissimilarity between activity patterns for individual digit movements, measured using the cross-validated Mahalanobis distance (*Nili et al., 2014*). By comparing all possible pairs of digit-specific activity patterns, we obtained the representational structure (*Figure 1D–E*). The resulting inter-digit dissimilarity values were averaged across digit pairs and participants within each group (*Figure 1B*). Small inter-digit dissimilarity indicates that voxels in the hand area are similarly activated across individual digits; larger dissimilarity implies individuated digit representation. In amputees, mean dissimilarity was

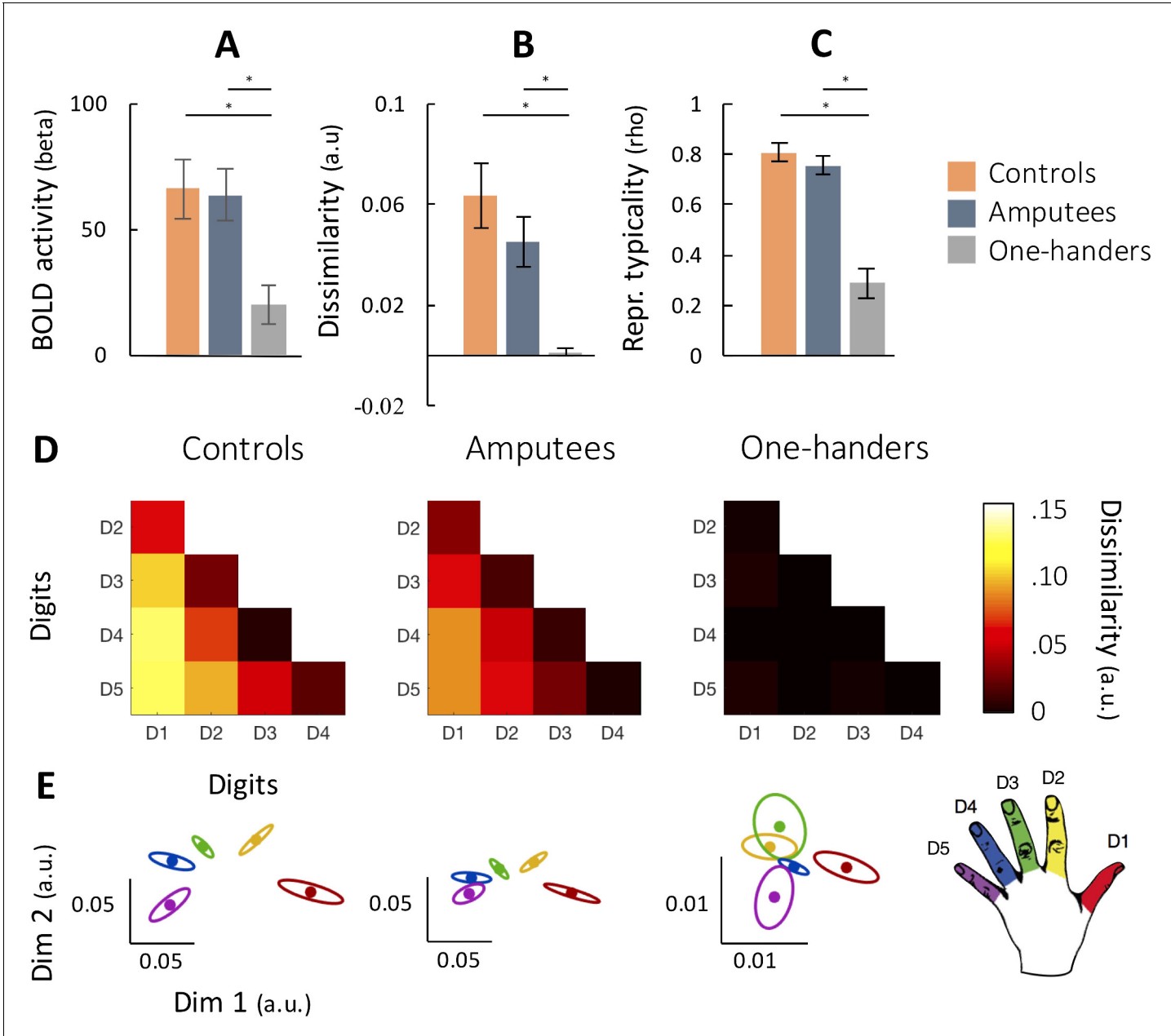

**Figure 1.** Similar representation in primary somatosensory cortex (SI) for amputees' missing hand and controls' nondominant hand, but not for congenital one-handers' missing hand. (**A**) Activity (averaged digit movement versus rest) in SI for amputees (n = 18), two-handed controls (n = 12), and congenital one-handers (n = 13). (**B–C**) Mean dissimilarity and typicality of the representational structure of contralateral SI activity for the three groups. (**D**) Representational dissimilarity matrices for the three groups. D1-D5 correspond to the five digits (thumb-little finger). (**E**) Two-dimensional projection of the representational structure (**D**) (using multi-dimensional scaling; note that this is included for visualisation purposes only and was not used for statistical analysis). Dissimilarity is reflected by distance in the two dimensions; individual digits are reflected by different colours (see colour key, bottom right); and ellipses reflect the between-subject standard error after Procrustes alignment. Please note the different scale for one-handers compared to amputees and controls. Abbreviations: a.u.: arbitrary unit; *: significant difference, after accounting for multiple comparisons.
DOI: https://doi.org/10.7554/eLife.37227.002

The following figure supplement is available for figure 1:

**Figure supplement 1.** Similar representation in primary motor cortex (M1) for amputees' missing hand and controls' nondominant hand, but not for the congenital one-handers' missing hand.
DOI: https://doi.org/10.7554/eLife.37227.003

slightly, though inconclusively, reduced compared to controls (t(28)=1.13, p=0.27, BF = 0.772), but significantly greater than in congenital one-handers (t(29)=3.54, p=0.001). Congenital one-handers showed no differentiation between digits of their missing hand (mean dissimilarity not different from 0; t(12)=.73, p=0.48) and dissimilarity was significantly reduced compared to controls (t(23)=4.86, p<0.001).

While the extent of discriminability is greater in amputees than congenital one-handers, it is possible that the pattern of individuated digit activity is atypical in amputees. To determine whether the inter-digit organisation of a missing hand is normal, we next studied the representational structure's typicality, that is the correlation of the representational dissimilarity matrix (RDM) with a dataset of hand RDMs in 2-handed controls drawn from a different study (*Wesselink et al., 2018*; see *Figure 1C* and Materials and methods). In amputees, on average, typicality was high (rho = 0.75) and was not significantly different from controls (t(28)=.991, p=0.33, BF = 0.128). Hence, the organisation of digit representation after hand-loss remained statistically unchanged, after an average of 18 years of handlessness. As expected, congenital one-handers' missing hand representation did not correlate with a normal hand pattern, reflected in diminished representational typicality (mean rho = 0.29) compared to both controls (t(23)=5.86, p<0.001) and amputees (t(29)=6.09, p<0.001).

The results in M1 were generally in line with our findings in SI, but, as expected (*Ejaz et al., 2015*; *Bruurmijn et al., 2017*), digit individuation was weaker (see *Figure 1—figure supplement 1*). Univariate activity in congenital one-handers' M1 was not significantly reduced compared to controls (t(23)=2.00, p=0.058) and greater than baseline (t(12)=4.60, p=0.001). Congenital one-handers showed significantly lower dissimilarity compared to controls (t(23)=4.23, p<0.001). In amputees, mean dissimilarity was slightly reduced compared to controls, but these differences were not significant (t(28)=.53, p=0.60, BF = 0.325). Typicality was also significantly lower in congenital one-handers than in either controls (t(23)=3.42, p=0.002) or amputees (t(29)=3.50, p=0.002), while the latter groups were not different from each other (t(28)=.11, p=0.91, BF = 0.253).

Although the inter-digit representational structure of congenital one-handers is atypical with respect to canonical hand representation, it is possible that it is still consistent within participants. To explore this idea, we split individual participants' data to odd and even scans. For each participant, we calculated an RDM in the missing/nondominant hand area using the odd and even runs, and correlated the two RDMs. The correlation between odd and even RDMs was significantly lower in congenital one-handers (rho = -.02) compared to both amputees (rho = 0.41; $p_{1H-AMP}$=.001) and controls (rho = 0.52; $p_{1H-CTR}$=.001). We note that by splitting the data we are reducing the effectiveness of our analysis. Nevertheless, the relative reduction in split-half consistency indicates that there is no strongly consistent digit information in the missing hand area of congenital one-handers during this task.

## Missing hand representation in acquired amputees is persistent even after phantom sensations have diminished

Next, we evaluated whether the consistency of hand representation in SI during missing hand movements correlates with amputees' subjective reports of phantom sensations. We first carried out an exploratory forward stepwise regression with typicality as the dependent variable. The following factors were tested as independent variables: kinaesthesia of phantom sensations - the number of phantom digits perceived as independently moving during the phantom movement task; vividness of nonpainful phantom sensations as experienced both during the study and chronically; intensity of phantom limb pain, as experienced both acutely during the study and chronically; time since amputation; age at amputation, and; typicality of the intact hand (calculated from the intact hand SI area). The final model (F = 19.9, p<0.001, adjusted $R^2$ = 0.645 included only kinaesthesia of phantom sensations (β = 0.07, t = 4.46, p<0.001) and the intercept (β = 0.52, t = 8.89, p<0.001). This regression was submitted to a bootstrapping analysis, allowing us to estimate the consistency of the final model (see Materials and methods). This bootstrapping analysis returned kinaesthesia as the final variable in 96.3% of the iterations (final model fit: median adjusted $R^2$ = 0.645; 95% CI: 30-99%). The proportion of the other included factors in the final model was: typicality of the intact hand (7.0%); time since amputation (9.5%); age at amputation (9.6%); vividness of nonpainful phantom sensations (acute: 12.2%; chronic: 20.8%); phantom limb pain, acute: 22.7%; chronic: 10.1%).

Post-hoc analysis confirmed a significant correlation between typicality and kinaesthesia in amputees (*Figure 2A*; rho = 0.72, df = 16, p=0.001). No significant correlation was found with nonpainful

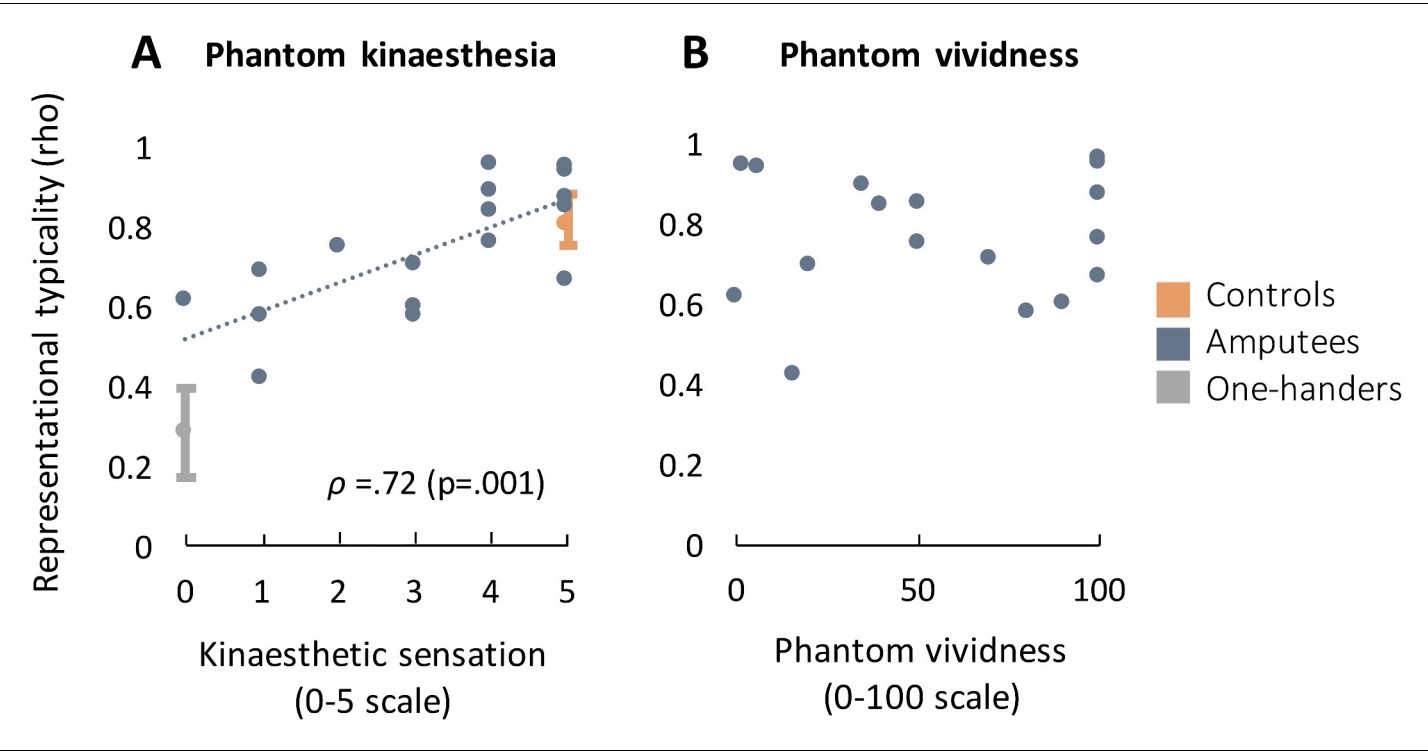

**Figure 2.** Kinaesthetic sensations during individuated phantom hand movements in amputees correlate with typicality in the missing hand's primary somatosensory cortex (SI). Typicality is the correlation coefficient of the representational dissimilarity matrix (RDM) with an independent hand RDM in controls. Phantom kinaesthesia (**A**) shows the number of digits that produced a sensation of movement during volitional phantom digit movements, based on amputees' self-reports. Grey and orange ranges show the mean and confidence intervals for typicality in one-handers and controls, respectively. The regression line is only presented for visualisation. Nonpainful phantom vividness (**B**) conveys the chronicity of the experience of the existence of a missing hand, where 0 indicates no sensations and 100 sensations identical to the intact hand.
DOI: https://doi.org/10.7554/eLife.37227.004

phantom vividness (the chronicity of experiencing the missing hand as existing; *Figure 2B*; rho = 0.13, df = 16, p=0.61), or years since amputation (rho = 0.17, df = 16, p=0.49). The correlation between kinaesthesia and dissimilarity in SI approached significance (rho = 0.447, p=0.063).

Further analysis confirmed that the correlation between kinaesthesia and typicality remained significant in SI after accounting for typicality in M1 (mean rho = 0.44) as a covariate (F = 17.73, p<0.001, Adjusted $R^2$ = 0.51; $\beta_{Kinaesthesia}$ =.74, p<0.001). This analysis suggests that although better recruitment of M1 is expected in individuals with clearer kinaesthetic sensations, the strong correlation between kinaesthesia and typicality in SI does not merely reflect information in M1.

Regardless of the positive relationship between kinaesthesia and typicality, amputees with little to no kinaesthetic sensations still showed missing hand representation. As stated above, the regression line between kinaesthesia and typicality had an intercept of $\beta_{intercept}$=.52. This was also the case when phantom vividness was the (non-significant) dependent variable (F = 0.021, p=0.89, Adjusted $R^2$ = -.061; $\beta_{intercept}$=.75, p<0.001). These results predict that even amputees who do not experience any phantom sensations will retain some typical missing hand representation. To test this prediction directly, we examined the three amputees in our dataset showing weak to no chronic phantom vividness (below 10/100). Despite not being able to experience clearly their phantom hand when performing the phantom movements task, these individuals showed high typicality (average typicality (rho) = 0.83). Moreover, when comparing their typicality to that found in the congenital one-handers (who were arguably better matched to this sub-group in terms of task demands), the amputees with diminished phantom sensations showed significantly stronger correlations with the canonical hand structure (Mann-Whitney U = 38, p=0.007). Typicality was not different between these three amputees and controls (Mann-Whitney U = 29, p=0.52, BF = 0.089). Together, these additional analyses

confirm that the representational structures' typicality in SI of amputees is still present in those with little to no phantom or kinaesthetic sensations.

## Diminished missing hand representation in congenital one-handers even when task performance is matched

While the task involving phantom hand movements was suitable to test the persistence of missing hand representation in individuals with phantom sensations, it was not designed to rule out the existence of missing hand representation in congenital one-handers. Indeed, it is possible that congenital one-handers have typical sensorimotor representation of their missing hand, but they did not access it due to unnatural task demands (see *Striem-Amit et al., 2015* for analogous results regarding visual cortex organisation in congenitally blind individuals).

To probe digit structure in the missing hand cortex using an alternative task, we examined whether we could observe a representation of the ipsilateral (intact) hand in the missing hand cortex. In two-handed controls, finger movements lead to individuated digit representation in specific cortical patches in ipsilateral M1 and SI, which tightly correspond to the activity patches engaged in the movement of the mirror-symmetric contralateral finger (*Diedrichsen et al., 2013*). Importantly, this ipsilateral digit representation fully overlaps with the representation of the contralateral hand (*Diedrichsen et al., 2018*). Furthermore, ipsilateral representation disappears completely during asymmetric bimanual finger movements, during which activity in M1 and SI is fully determined by the contralateral hand (*Diedrichsen et al., 2013*). As such, the ipsilateral representation of one hand is likely elicited due to recruitment of the representation of the contralateral hand (*Diedrichsen et al., 2018*; *Berlot et al., 2018*). Ipsilateral representation of the intact hand can therefore provide an indirect assay into the representation of the missing hand, while controlling for task demands across groups. Importantly, all three groups were able to perform the individuated digit movement task equally well and contralateral representation of the intact hand was typical in all groups (see Materials and methods). We compared the intact/dominant inter-digit representational structure in the missing/nondominant hand area of one-handers/controls (respectively). We predicted that persistent missing hand representation in amputees should result in similar ipsilateral representation in their missing hand cortex as controls. If missing hand representation is diminished in congenital one-handers, then ipsilateral representation of their intact hand (in the missing hand area) should show reduced representational features compared to those found in amputees (see Discussion for an alternative mechanism, where the deprived cortex develops separate representations for both the contralateral (missing) and ipsilateral (intact) hands).

Mean (intact hand) ipsilateral digit dissimilarity and typicality were not significantly different between amputees and controls (*Figure 3*; dissimilarity: $t_{(28)}$=1.42, p=0.166, BF = 0.209; typicality: $t_{(28)}$=.69, p=0.498, BF = 0.244). In contrast, ipsilateral representation in congenital one-handers was significantly lower than in amputees (dissimilarity: $t_{(29)}$=3.81, p<0.001; typicality: $t_{(29)}$=3.05, p=0.005) and showed similar trends versus controls (dissimilarity: $t_{(23)}$=2.20, p=0.038; typicality: $t_{(23)}$=2.19, p=0.039). Together, these findings provide additional support for the reduced existence of inter-digit representational difference in the missing hand cortex of congenital one-handers versus amputees, independent of missing hand motor skill.

It is important to consider that we found some potential support for the existence of an ipsilateral digit representation in the missing hand cortex of congenital one-handers. The cross-validated dissimilarity measurement in this group was significantly larger than zero (t(12)=2.51, p=0.027), indicating that there were significant differences between the activity patterns associated with each finger. We therefore wished to determine whether the dissimilarity measures reflect meaningful (though reduced) sensorimotor digit information, or rather the increased sensitivity of RSA to other (not sensorimotor) inter-digit differences (e.g. visual task differences). For example, a recent study demonstrated that visual information about touch on the hand is sufficient to induce some residual digit-selective activity patterns in SI (*Kuehn et al., 2018*). We therefore compared representational measurements between SI and visual area V5, previously shown not to contain individual sensorimotor digit representation (*Beauchamp et al., 2009*). Although it is difficult to set a benchmark at a particular dissimilarity value, we suggest that representation crucial to somatosensation in SI should at least outperform V5. Congenital one-handers showed no significant differences in representation between SI and V5 (dissimilarity: t(12)=-1.03, p=0.322; typicality: t(12)=.78, p=0.448), whereas both amputees and controls showed significant differences (all p's < 0.02), resulting in a significant group

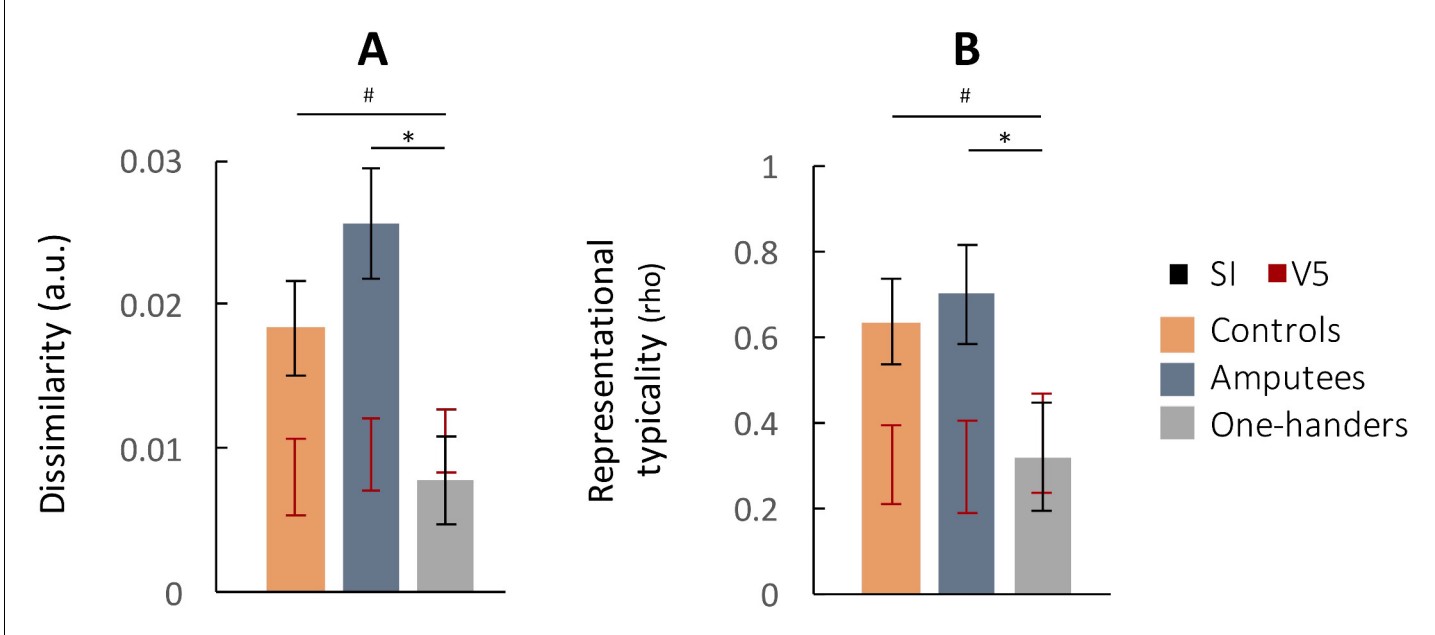

**Figure 3.** Similar ipsilateral hand representation in primary somatosensory cortex (SI) for amputees' and controls' intact hand. (A–B) Mean dissimilarity and typicality of the representational structure of ipsilateral SI activity for the three groups. Both dissimilarity and typicality of ipsilateral hand representation indicate a difference between missing hand representation in congenital one-handers and amputees, independent of missing hand motor skill. The red error bars indicate the dissimilarity and typicality values (standard error of the mean) in a visual control area V5 for the same groups, designed to capture visuomotor representation that is not strictly somatosensory. While amputees and controls showed significantly greater digit representation in SI than V5 (both in terms of dissimilarity and typicality), congenital one-handers did not, further indicating reduced SI digit representation. Abbreviations: a.u.: arbitrary unit; *: significant difference; #: trending difference (.02 < p < 0.05).

DOI: https://doi.org/10.7554/eLife.37227.005

x area interaction (dissimilarity: $F_{(2,85)}=4.25$, p=0.018; typicality: $F_{(2,85)}=4.42$, p=0.015; *Figure 3*). These findings corroborate that the digit dissimilarity values in congenital one-handers are likely not specifically indicative of sensorimotor representation, but may rather reflect other task demands.

## Discussion

Here, we demonstrated long-term stability of SI hand representation in a group of acquired amputees with diverse phantom sensations, including those experiencing limited phantom sensations. Using RSA, we find that amputees show individuated digit representation for their missing hand, as exemplified by significantly greater inter-digit dissimilarity values in amputees versus congenital one-handers. The inter-digit pattern comprising the missing hand representation was typical to SI hand representation in amputees and was not significantly different from controls (as supported by a Bayesian analysis). Importantly, by studying individuals with a varying range of phantom sensations, we were able to confirm stable hand representation even after phantom sensations have diminished. This result confirms the persistence of hand representation as a general principle in amputees, contrary to recent reports (*Serino et al., 2017*). Using the same task, we were unable to identify similar digit representation for the missing hand of congenital one-handers, demonstrated by significantly reduced pattern typicality compared to amputees (and even in comparison to those few amputees with little phantom sensations), as well as controls. This result confirms that the persistent hand representation observed in amputees does not reflect mere cognitive task demands (e.g. visual feedback, *Kuehn et al., 2018*; or attention, *Puckett et al., 2017*).

We also explored whether we could activate the representation of the missing hand indirectly by movements of the fingers of the intact hand. Previous studies in two-handers have demonstrated that contralateral and ipsilateral hand movements produce identical representational patterns (*Diedrichsen et al., 2018*). Since this ipsilateral representation is completely overwritten by the

contralateral hand if the two hands are engaged in dissociated movements (*Diedrichsen et al., 2013*, Exp. 2), it has been proposed that the ipsilateral hand reactivates the cortical resources associated with the contralateral hand. Based on this evidence, the ipsilateral digit representation can serve as an indirect measure of the contralateral hand representation. This discovery provides us with a unique and novel opportunity to interrogate the information content underlying the 'deprived' hand cortex despite the physical absence of the hand. This approach provided converging evidence, but using different task demands, for similar (missing) hand representation for amputees and controls, but not for congenital one-handers. It might be worth considering whether congenital one-hander's deprived cortex could have developed separate representations for both the contralateral (missing) and ipsilateral (intact) hand, which are uncorrelated due to anatomical or behavioural differences from two-handers. If this were possible, we would predict maintained, or even stronger, ipsilateral representation of the intact hand. Yet, our data does not reflect this hypothesis. Moreover, poor ipsilateral representation in the deprived cortex does not seem to stem from reduced inter-hemispheric connectivity, which appears to be functionally and structurally preserved in congenital one-handers, depending on lateralisation strategies in daily behaviour (*Hahamy et al., 2015*; *Hahamy et al., 2017*; *Makin et al., 2013a*). Finally, it is still possible that rudimentary missing hand representation, for example determined by genetic factors (*Miyashita-Lin et al., 1999*; *Rubenstein et al., 1999*), has originally formed in congenital one-handers but later diminished due to lack of consistent sensorimotor input. Bearing this caveat in mind, our findings suggest that early-life experience is potentially necessary to create typical functional sensory organisation, but not to maintain it.

It has previously been shown that restored peripheral input, for example via hand transplantation (*Frey et al., 2008*) or targeted reinnervation (*Serino et al., 2017*) can reinstate sensorimotor hand representation, indicating that the canonical hand representation is, to an extent, immutable to change. Moreover, we reported that the SI hand map can persist independently of the original peripheral input, as observed in a patient sustaining a brachial plexus avulsion injury, abolishing communication between the periphery and the central nervous system (*Kikkert et al., 2016*). Similarly, in the current study amputees showed, on average, persistent SI representation despite suffering diverse nerve injuries spanned varying degrees of amputation (*Table 1*). It is therefore necessary to consider alternative inputs that might contribute to the stability of the missing hand map in amputees. Considering that our task required active phantom movements, it is likely that the SI representation pattern is driven by motor efferent inputs. Indeed, while motor signals can no longer reach their final output muscle terminal, the motor cortex in amputees remains functional (*Raffin et al., 2012a*; see *Kokotilo et al., 2009* for similar results in spinal cord injury patients). When a motor command is sent out (e.g. in the form of an attempted hand movement), efference signals are thought to reach SI and generate corollary discharge, suggested to resemble the expected sensory feedback activity pattern, resulting from the movement (*London and Miller, 2013*; *Adams et al., 2013*). Since congenital one-handers have never operated a hand, it is likely that this sensorimotor predictive coding architecture never formed in the first place, explaining the lack of inter-digit dissimilarity found in the present study.

It is important to consider how the finding of robust persistence of hand representation, despite the physical absence of a hand, conceptually aligns with other reports of brain reorganisation. Since the pioneering work of Hubel and Wiesel, demonstrating that input loss to visual cortex in early development leads to profound physiological changes (*Wiesel and Hubel, 1965b*; *Hubel and Wiesel, 1965*; *Wiesel and Hubel, 1965a*), it has long been established that sensory deprivation causes cortical reorganisation. Later seminal electrophysiological studies in monkeys further demonstrated that deprivation-driven reorganisation following peripheral input-loss also occurs in adults (*Kaas et al., 1983*). For example, following peripheral deafferentation of the hand and arm, the missing hand SI area becomes responsive to touch applied to the monkey's lower face (*Pons et al., 1991*), likely due to subcortical re-routing of inputs (*Kambi et al., 2014*; *Liao et al., 2016*). Recent research in humans indicates extensive reorganisation of multiple body-part representations onto the deprived hand area of congenital one-handers (*Hahamy et al., 2017*; *Striem-Amit et al., 2018*; *Stoeckel et al., 2009*). In amputees sustaining input loss in adulthood, original reports emphasised facial remapping in SI, akin to the reorganisation observed in monkeys (*Flor et al., 1995*), as a driving mechanism for phantom limb pain (maladaptive plasticity; *Flor et al., 2006*). Later research challenged the notion that the deprived hand area gets taken over by facial inputs (*Makin et al., 2013b*;

**Table 1.** Summary demographic details and phantom sensations.

Data is shown for amputees (AMP), controls (CTR) and congenital one-handers (1H). Congenital one-handers did not feel any phantom limb sensations. All controls have full kinaesthetic sensations. F: female, M: male. Side: side of missing hand; L: left, R: right. Amputation level: 1: shoulder, 2: above elbow; 3: at elbow; 4: below elbow; 5: at wrist. Kin: Phantom limb kinaesthesia (number of independent controllable parts of the hand), Viv: Chronic phantom limb vividness (0: no sensation, 100: intact hand's vividness), Pain: Chronic phantom limb pain (0: no pain, 100: worst pain imaginable), AViv/APain: Acute Viv/Pain (on the scanning day), Std: standard deviation, ND: nondominant.

| AMP | Age (years) | Sex | Side | Years since | Age at (years) | Level | Kin (0-5) | Viv (0-100) | Pain (0-100) | AViv (0-100) | APain (0-100) |
|---|---|---|---|---|---|---|---|---|---|---|---|
| | | | | **Amputation** | | | | **Phantom sensations** | | | |
| Mean | 50.4 | | | 17.6 | 32.9 | | 3 | 58 | 46 | 65 | 21 |
| St. dev. | 12.1 | | | 10.4 | 11.8 | | 2 | 38 | 37 | 30 | 23 |
| A01 | 44 | M | R | 15 | 29 | 2 | 5 | 100 | 100 | 100 | 50 |
| A02 | 53 | M | L | 32 | 21 | 2 | 5 | 50 | 100 | 60 | 70 |
| A03 | 40 | M | L | 11 | 29 | 2 | 4 | 100 | 50 | 100 | 20 |
| A04 | 51 | M | L | 32 | 19 | 2 | 5 | 100 | 0 | 100 | 0 |
| A05 | 27 | F | R | 7 | 20 | 4 | 2 | 50 | 40 | 60 | 0 |
| A06 | 71 | M | R | 16 | 55 | 2 | 1 | 20 | 85 | 60 | 20 |
| A07 | 46 | M | R | 18 | 28 | 2 | 3 | 70 | 90 | 70 | 50 |
| A08 | 56 | M | L | 26 | 30 | 4 | 5 | 6 | 40 | 10 | 0 |
| A09 | 64 | M | L | 31 | 33 | 2 | 4 | 100 | 40 | 100 | 10 |
| A10 | 58 | M | L | 2 | 56 | 2 | 3 | 90 | 0 | 80 | 0 |
| A11 | 28 | M | L | 8 | 20 | 5 | 4 | 40 | 40 | 20 | 0 |
| A12 | 57 | M | R | 29 | 28 | 2 | 1 | 80 | 90 | 80 | 40 |
| A13 | 50 | F | L | 1 | 49 | 4 | 0 | 0 | 0 | 0 | 0 |
| A14 | 52 | M | R | 27 | 25 | 2 | 5 | 100 | 80 | 80 | 50 |
| A15 | 68 | M | R | 26 | 42 | 4 | 1 | 16 | 0 | 80 | 0 |
| A16 | 39 | F | R | 9 | 30 | 3 | 4 | 35 | 40 | 50 | 30 |
| A17 | 58 | M | L | 12 | 46 | 4 | 5 | 2 | 0 | 65 | 0 |
| A18 | 46 | F | L | 14 | 32 | 4 | 3 | 80 | 30 | 50 | 30 |

| CTR | Age (years) | Sex | Side (ND hand) | | 1H | Age (years) | Sex | Side (Missing hand) | Level |
|---|---|---|---|---|---|---|---|---|---|
| Mean | 45.3 | | | | Mean | 45.7 | | | |
| St. dev. | 14.9 | | | | St. dev. | 10.4 | | | |
| C01 | 29 | M | R | | H01 | 41 | M | L | 4 |
| C02 | 24 | F | L | | H02 | 37 | M | R | 4 |
| C03 | 47 | F | L | | H03 | 31 | F | L | 4 |
| C04 | 39 | M | L | | H04 | 60 | M | L | 4 |
| C05 | 32 | M | R | | H05 | 39 | F | L | 4 |
| C06 | 53 | F | R | | H06 | 54 | F | L | 4 |
| C07 | 38 | F | R | | H07 | 34 | M | L | 4 |
| C08 | 67 | M | R | | H08 | 63 | M | L | 4 |
| C09 | 42 | M | R | | H09 | 44 | F | R | 4 |
| C10 | 41 | M | R | | H10 | 55 | F | L | 4 |
| C11 | 69 | M | L | | H11 | 46 | M | R | 4 |
| C12 | 63 | F | L | | H12 | 37 | M | R | 4 |
| | | | | | H13 | 53 | F | L | 4 |

DOI: https://doi.org/10.7554/eLife.37227.006

*Makin et al., 2015*; *Raffin et al., 2016*) and instead emphasised increased representation of the intact hand in the missing hand cortex as a potential neural correlate of adaptive plasticity (*Makin et al., 2013a*; *Philip and Frey, 2014*; see further discussion below). More recently, we have suggested that functional reorganisation is more limited than originally considered (*Makin and Bensmaia, 2017*). Regardless of the ongoing debate over the functional role of SI reorganisation in the adult (*Andoh et al., 2018*; *Kuner and Flor, 2017*) and developing brain (*Hahamy et al., 2017*; *Striem-Amit et al., 2018*), common to all these previous studies of reorganisation is that to activate the deprived cortex researchers studied representations of the spared body parts (e.g. the face or the intact hand). While this approach is suitable for documenting cortical remapping, it leaves unexplored the possibility that the original functional organisation of the now-deprived area may be preserved, though latent. We propose that reorganisation in the missing hand cortex does not necessarily abolish the original functional layout in sensory cortex. For example, persistent representation, in the form of efferent cortico-cortical input would engage a separate cortical layer (*Felleman and Van Essen, 1991*; *Adams et al., 2013*) than brainstem and thalamic facial inputs to the deprived cortex (*Kambi et al., 2014*). It still remains to be determined whether these two forms of persistent representation and reorganisation are functionally orthogonal, or interactive (*Andoh et al., 2018*).

As mentioned above, the missing hand cortex in amputees, but not in congenital one-handers, has been previously shown to respond to inputs from the intact hand (*Bogdanov et al., 2012*; *Makin et al., 2013a*; *Hahamy et al., 2017*; *Philip and Frey, 2014*), presumably through functional reorganisation. Here, we used RSA to dissect the information content underlying ipsilateral activity of the intact hand. We find that amputees, but not congenital one-handers, showed similar measures of dissimilarity and typicality as controls. However, the fact that ipsilateral dissimilarity was not significantly greater in amputees than in controls is inconsistent with the interpretation of increased intact hand activity as a neural correlate of adaptive reorganisation (*Makin et al., 2013a*). Regardless, the existence of ipsilateral digit-specific organisation in the missing hand cortex of amputees might provide an alternative mechanism for the preservation of the missing hand digit maps. While we previously showed that the phantom hand map is activated by phantom hand movements independently of the intact hand (*Kikkert et al., 2016*; *Philip and Frey, 2014*), it is still possible that structured inputs from the intact hand (via ipsilateral pathways) sustains the missing hand map, despite the loss of the original peripheral inputs.

To conclude, here we show that once sensorimotor hand-representation is formed, it is generally immutable to change: We identified stable hand representation in amputees' sensorimotor cortex using representational similarity analysis, despite years (and even decades) of amputation and irrespective of their phantom sensations vividness. In contrast, individuals born with a missing hand (congenital one-handers) did not show normal representation of their nonexisting hand. We therefore suggest that consistent sensory representation despite input loss may be a common organising principle (*Striem-Amit et al., 2015*; *Collignon et al., 2013*; *Baseler et al., 2011*). How can our findings of persistent representation, despite massive and long-lasting input change, be resolved with multiple observations of updated hand representation due to altered experience (e.g. due to nerve/digit deafferentation [*Merzenich et al., 1983*; *Merzenich et al., 1984*], increased usage [*Jenkins et al., 1990*], syndactyly [*Allard et al., 1991*; *Wang et al., 1995*], or mobile phone usage [*Gindrat et al., 2015*])? Here we show that amputees with greater phantom kinaesthetic sensations better retained their missing hand representation. In light of this, we suggest that daily life experience could shape the fine-grained aspects of hand representation, but the large-scale functional organisation of the hand area is fundamentally stable.

## Materials and methods

### Participants

We tested 18 acquired amputees with an average of 18 years since amputation (mean age: 50 ± 12; eight left-handed; four female), 13 congenital one-handers (mean age: 46 ± 10; four left-handers; six female), and 12 two-handed control participants (mean age: 45 ± 15; five left-handers; five female). All amputees reported experiencing phantom sensations after amputation, but vividness of these sensations varied across participants at the time of the study (mean chronic vividness score 58 ± 38

on a 0–100 scale, as assessed using questionnaires [*Makin et al., 2013b*; *Makin et al., 2015*; *Makin et al., 2013a*; see *Table 1* and Questionnaires section below for further details]). Three participants in the amputees group tested here also took part in our previous study (*Kikkert et al., 2016*). The congenital one-handers had never experienced any phantom sensations. In addition, we also recruited and excluded a further congenital one-hander (due to technical difficulties during data pre-processing) and two control participants (due to incomplete data collection and due to abnormal digit selectivity, i.e. more than three standard deviations from the mean).

Recruitment was carried out in accordance with the University of Oxford's Medical Sciences interdivisional research ethics committee (MS-IDREC-C2-2015-012). Informed consent and consent to publish was obtained in accordance with ethical standards set out by the Declaration of Helsinki. Control participants were recruited as to match the other two groups in term of age, gender and handedness (with respect to the intact hand). When possible, control participants were friends and family of the one-handed participants. All participants were compatible with local magnetic resonance imaging (MRI) safety guidelines.

## Experimental procedures

The experimental procedures described in this manuscript were run as part of a larger study (the full study protocol can be found on https://osf.io/gmvua/). Here we focus on procedures related to the representation of the missing hand in amputees and congenital one-handers.

## Questionnaires

To measure phantom sensations, as well as other demographic and clinical details of potential relevance to the missing hand representation, amputees and congenital one-handers completed a range of questionnaires (as summarised in *Table 1*). Amputees rated intensities of phantom sensations, using a 0–100 scale, as experienced during the last week (or in a typical week involving such sensations). Chronic phantom sensation was calculated by dividing intensity by sensations frequency (1- all the time; 2- daily; 3- weekly; 4- several times per month; and 5- once or less per month), as previously implemented (*Makin et al., 2013b*; *Makin et al., 2015*). Having used this measure in multiple studies with partly overlapping participant pools (*Makin et al., 2013b*; *Kikkert et al., 2016*; *van den Heiligenberg et al., 2017*) we can assess the consistency of this measure within participants and across studies (i.e. measure reliability). We found excellent inter-study consistency (intra-class correlation coefficient: 0.79, 95% CI: .48-.93, F(13,13)=8.46, p<0.001), when considering all amputees that participated in at least one other study (n = 14, earlier questionnaire taken 1–4 years before current study). In addition, participants reported the number of phantom digits that afford kinaesthetic sensations during volitional control of movements (kinaesthesia). This report was further validated by a demonstration of afforded phantom movements during the study's main task with the intact hand, as detailed below.

## MRI tasks

All participants underwent one experimental session with four fMRI runs, using a block-design. The task involved individual digit-movement blocks for each of the five digits (12 s blocks) of either hand, as well as no movement (rest) blocks. Each condition was repeated three times in a semi-counterbalanced order within each run. Each run comprised a different block order.

To probe somatosensory digit representation, we used a visually cued active (motor) task. In an intact sensorimotor system, movement recruits a combination of peripheral receptors, encoding a range of somatosensory modalities (e.g. surface and deeper mechanoreceptors; proprioceptors), as well as efferent information from the motor system. Using an active task, we have previously shown high consistency of SI digit topography across multiple scanning sessions (*Kolasinski et al., 2016*, see also *Ejaz et al., 2015* for validation using RSA). Participants were presented with five vertical bars, corresponding to the five digits, shown on a visual display projected into the scanner bore. To cue the participant which digit should be moved, the bar corresponding to this digit changed (i.e. by flashing in a different colour).

On 'missing hand blocks', participants were instructed to perform individual digit movements (1 Hz) with their nondominant (controls), phantom (amputees), or missing hand (congenital one-handers). Handless individuals were instructed to attempt performing actual movements with the digits of

their missing hand, even when not being able to feel their digits, rather than using motor imagery. Controls moved their nondominant hand digits in mid-air. To ensure good understanding of these instructions, outside the scanner, the amputees were asked to demonstrate to the experimenter the extent of volitional movement they felt they were able to carry out in each of their phantom digits, by mirroring each movement onto their intact hand.

On 'intact hand blocks', all participants performed a comparable task with their intact/dominant hand by exerting force on a button box. Participants received real-time visual feedback of how much force each digit exerted by means of moving vertical bars on 'intact hand blocks', but not on 'missing hand blocks'. The dominant hand of controls was paired up with the intact hand because, through intensive use, amputees' and congenital one-handers' intact hand becomes their de facto dominant hand (*Philip and Frey, 2014*). All groups were able to carry this task equally well, as verified in post-hoc analysis: each trial was assigned to the digit whose force output correlated most strongly with the instructed time course and the percentage of correctly performed trials, that is trials that were assigned to the instructed digit, was not different between congenital one-handers and amputees (74.2%, $t(29)=1.13$, $p=0.266$), or between congenital one-handers (81.1%) and controls (75.3%; $t(23)=.93$, $p=0.362$). This behaviour brought forth high representational typicality in intact SI for all three groups (controls: rho = 0.85; amputees: rho = 0.81; and congenital one-handers: rho = 0.89; group comparisons all p's > 0.11).

## MRI acquisition

MRI images were acquired using a 3T MAGNETON Prisma MRI scanner (Siemens, Erlangen, Germany) with a 32-channel head coil. Functional images were collected using a multiband T2*-weighted pulse sequence with a between-slice acceleration factor of 4 and no in-slice acceleration. This provided the opportunity to acquire data with high spatial (2 mm isotropic) and temporal (TR: 1500 ms) resolution, covering the entire brain. The following acquisition parameters were used: TE: 32.40 ms; flip angle: 75°, 72 transversal slices. Field maps were acquired for field unwarping. A T1-weighted sequence was used to acquire an anatomical image (TR: 1900 ms, TE: 3.97 ms, flip angle: 8°, spatial resolution: 1 mm isotropic).

## MRI analysis

MRI analysis was implemented using tools from FSL, SPM and Connectome Workbench software (*Smith et al., 2004*; *Jenkinson et al., 2012*, https://fsl.fmrib.ox.ac.uk/fsl/fslwiki/, https://www.fil.ion.ucl.ac.uk/spm/, humanconnectome.org) in combination with other Matlab scripts (version R2016a), both developed in-house (*Wesselink and Maimon-Mor, 2017*) and as part of the RSA Toolbox (*Nili et al., 2014*). Cortical surface reconstructions were produced using FreeSurfer (*Dale et al., 1999*; *Fischl et al., 2001*, freesurfer.net).

### fMRI pre-processing

Functional data was first pre-processed in FSL 5.0. The following steps were included: Motion correction using MCFLIRT (*Jenkinson et al., 2002*), brain extraction using BET (*Smith, 2002*), and high pass temporal filtering with a cut-off of 100 s. Co-registration to each individual anatomical T1 scan was accomplished using FLIRT and, where needed, manual adjustments were performed to ensure precise co-registration around the hand knob of the central sulcus.

Anatomical T1 images were used to reconstruct the pial and white-grey matter surfaces using Freesurfer. Surface co-registration across hemispheres and participants was done using spherical alignment. Individual surfaces were nonlinearly fitted to a template surface, first in terms of the sulcal depth map, and then in terms of the local curvature, resulting in a nearly perfect overlap of the fundus of the central sulcus across participants (*Fischl et al., 2008*).

### Regions of Interest (ROI) definition

Since the focus of the study was on persistent sensory representation, our main analysis was restricted to the individualised hand-selective ROIs in SI. Further analysis was focused on the M1 hand areas. The ROIs were always in the hemisphere contralateral to the missing/nondominant hand. The anatomical ROIs were defined on the group surface using probabilistic cytotectonic maps aligned to the average surface (see *Wiestler and Diedrichsen, 2013*). These regions were then

projected into the individual brains via the reconstructed individual anatomical surfaces. For the hand area of SI, we selected all surface nodes with the highest probability for any of BA3a, 3b, 1, and 2, surrounding the anatomical hand knob (*Yousry et al., 1997*). The hand area of M1 was selected similarly using BA 4. We note that given the probabilistic nature of these masks, the dissociation between SI and M1 is only an estimate. For one acquired amputee, surface alignment failed; for this subject, the ROIs were drawn manually within the surface ribbon using the above anatomical definitions. The ROIs were not significantly different in size across groups (one-way ANOVA on area volume: SI: F = 1.27, p=0.29; M1: F = 1.37, p=0.27). In addition, for control purposes, we also used an ROI of visual area V5 which we defined anatomically, based on the parameters previously published by *Wiestler and Diedrichsen (2013)*. The ROI was constructed bilaterally and RSA outcome measures were averaged across both hemispheres.

## fMRI analysis

Voxel-wise General Linear Model (GLM) analysis was carried out, as implemented in SPM12. In brief, each of the experimental conditions was modelled for each run separately against rest. Regressors were created by convolving stimulus presentation (as a boxcar function) with a double-gamma hemodynamic response function (HRF). In the GLM estimation, the functional data was weighted using the robust Weighted Least Squares approach (*Diedrichsen and Shadmehr, 2005*), which estimates the heteroscedasticity of the time series and then 'soft'-excludes noisy image volumes (e.g. due to movement). Task-related activity was quantified by averaging the BOLD response, averaged across all digits, versus baseline within each ROI. The voxel-wise parameter estimates (hereafter: activity patterns) and residuals from this analysis were also used to calculate the dissimilarity, as detailed below.

The dissimilarity between activity patterns within each ROI was measured for each digit pair using the cross-validated squared Mahalanobis distance, or 'crossnobis' distance (*Nili et al., 2014*). We calculated the distances using each possible pair of imaging runs and then averaged the resulting distances. Before estimating the dissimilarity for each digit pair, the activity patterns were pre-whitened using the residuals from the GLM. Due to cross-validation, the expected value of the distance is zero (but can go below 0) if two patterns are not statistically different from each other, and larger than zero if there is differentiation between the digits of the hand.

We extracted two measures from the resulting inter-digit representational dissimilarity matrix (RDM). As a measure of strength of the representation, we used the mean dissimilarity, the average dissimilarity between the ten unique digit pairs (excluding the diagonal). The typicality of the representational structure was assessed by calculating the Spearman's rho correlation between the measured RDM and the average RDM of the dominant hand of two-handed controls (independently acquired; see below). Because the representational structure can be related to behavioural aspects of hand use and is highly invariant in controls (average correlation r = 0.9, *Ejaz et al., 2015*), this measure serves as a proxy for how 'normal' the hand representation is. Being able to study this measure was a main reason for using RSA in this study.

As an aid to visualise the RDMs, we also used classical multidimensional scaling (MDS). MDS projects the higher-dimensional RDM into a lower-dimensional space, while preserving the inter-digit dissimilarity values as well as possible (*Borg and Groenen, 2005*). MDS was performed on data from individual participants and averaged after Procrustes alignment to remove arbitrary rotation induced by MDS. Note that MDS is presented for intuitive visualisation purposes only, and was not used for statistical analysis.

As mentioned above, to determine typicality we correlated RDM from the current study with the average representational structure of the dominant hand of two-handed controls, in an independently acquired cohort of participants. The full details of the acquisition parameters are described in *Wesselink and Maimon-Mor, 2017*. In short, eight two-handed participants performed an active digit tapping task using a button box (four repetitions per digit of 8 s blocks of 1 Hz single-digit presses), without online visual feedback. The data was acquired at 7T (TR: 2000 ms, TE: 25 ms, voxel size: 1 × 1×1 mm). The ROI was defined similarly to the SI ROI used in the current study.

## Statistical analysis

Statistics were calculated using Matlab R2016a. Subsequent to normality validation (using the Sha-piro-Wilk test), we used paired/independent-sample two-tailed Student's t-tests to compare activity levels and distance measures within/between groups, and one-sample t-tests to compare group measures to zero. Correlations were calculated as Spearman's rho. Partial correlation effects were calculated using linear regression. For the main analysis concerning RSA, within each ROI, group t-tests were adjusted to three comparisons, using the Bonferroni correction ($\alpha = 0.05/3$), to account for the three inter-group comparisons. Post-hoc correlations between SI typicality and key clinical measurements (phantom sensations, phantom kinaesthesia and time since amputation) were also corrected for three comparisons ($\alpha = 0.05/3$). One post-hoc comparison involving a small subset of amputees was done using a Mann-Whitney U test (see Results - section 2). For a control analysis, involving visual and somatosensory ROIs, we also used a mixed-design analysis of variance (ANOVA) to identify interactions across groups and ROIs.

In order to assess whether any aspect of the representational structure in the amputees was not different from that in controls, we used Bayesian statistics as implemented in Javascript (*Dienes, 2014*; *Singh, 2018*). Our alternative hypothesis is that amputees have no preserved hand representation. To construct our prior (i.e. to quantify the effect of having hand representation) we calculated the effect size of controls' nondominant hand representation vs. congenital one-handers missing hand representation. We then compared the effect size of amputees' missing hand representation (compared to controls) against that prior. More specifically, our alternative hypothesis assumes an effect size following a one-tailed t-distribution centred at 0 and a width of the difference between congenital one-handers and controls. The measured difference between amputees and controls (also modelled as a t-distribution) are tested against this hypothesis. Support for the null hypothesis was interpreted as supporting preserved hand representation. While it is generally agreed that it is difficult to establish a cut-off for what consists sufficient evidence, we used the threshold of BF<1/3 as positive evidence in support of the null, consistent with others in the field regarding this threshold as providing substantial evidence (*Wetzels et al., 2011*; *Dienes, 2014*). Note, however, that this threshold is not considered as providing strong evidence by all accounts (*Kass and Raftery, 1995*).

In order to gauge which aspects of phantom sensation and key demographics may relate to the amputees' representational structure's typicality, we performed an exploratory forward stepwise regression. The dependent variable was SI typicality in amputees. The following factors were used as independent variables (see also *Table 1*): typicality of the intact hand (rho; calculated from the intact hand SI area; time since amputation (in years); age at amputation (in years); vividness of nonpainful phantom sensations, as experienced during the study (on a 0–100 scale) and chronically (accounting for both intensity and frequency; *Makin et al., 2013b*); intensity of phantom limb pain, as experienced acutely during the study, and chronically (as detailed for nonpainful sensations). Only linear factors were considered, that is no interaction terms, and the criterion for inclusion was an increase in $R^2 > 0.1$. As a large number of predictor variables were included in the model and stepwise regression is generally only recommended for exploratory analysis, we aimed to establish internal replicability using bootstrap resampling (e.g. *Thompson, 1995*). In particular, we randomly sampled (with replacement) the full data matrix and repeated the stepwise regression 1000 times. We subsequently computed the proportion of bootstrap samples in which each factor was included in the final model, as well as confidence bounds on the model's adjusted $R^2$. We interpreted high proportion of inclusion (p>0.75) as evidence for internal replicability (*Thompson, 1995*).

## Acknowledgements

This work was supported by a Sir Henry Dale Fellowship jointly funded by the Wellcome Trust and the Royal Society (104128/Z/14/Z), awarded to TRM. LC was supported by a CREATE-IRTG grant. We thank our participants for taking part in the study and Opcare for invaluable help with partici-pants' recruitment. We thank Laurie Josephs, Paulina Kieliba, Gonzague de France and Liezel Wegner for their help with data collection. We thank Devin Terhune for providing us with a consis-tency analysis for the step-wise regression.

## Additional information

### Competing interests

Jörn Diedrichsen, Tamar R Makin: Member of eLife BRE. The other authors declare that no competing interests exist.

### Funding

| Funder | Grant reference number | Author |
|---|---|---|
| Wellcome | 104128/Z/14/Z | Tamar R Makin |
| Royal Society | 104128/Z/14/Z | Tamar R Makin |
| CREATE-IRTG | | Lucilla Cardinali |

The funders had no role in study design, data collection and interpretation, or the decision to submit the work for publication.

### Author contributions

Daan B Wesselink, Data curation, Software, Formal analysis, Validation, Investigation, Visualization, Methodology, Writing—original draft, Project administration, Writing—review and editing; Fiona MZ van den Heiligenberg, Conceptualization, Data curation, Formal analysis, Investigation, Methodology, Project administration, Writing—review and editing; Naveed Ejaz, Conceptualization, Resources, Software, Formal analysis, Validation, Methodology, Writing—review and editing; Harriet Dempsey-Jones, Investigation, Project administration, Writing—review and editing; Lucilla Cardinali, Investigation, Writing—review and editing; Aurelie Tarall-Jozwiak, Resources, Writing—review and editing; Jörn Diedrichsen, Conceptualization, Software, Supervision; Tamar R Makin, Conceptualization, Resources, Data curation, Software, Formal analysis, Supervision, Funding acquisition, Validation, Investigation, Visualization, Methodology, Writing—original draft, Writing—review and editing

### Author ORCIDs

Daan B Wesselink http://orcid.org/0000-0002-3229-0925
Naveed Ejaz http://orcid.org/0000-0001-8370-4588
Lucilla Cardinali http://orcid.org/0000-0002-0441-1806
Jörn Diedrichsen https://orcid.org/0000-0003-0264-8532
Tamar R Makin http://orcid.org/0000-0002-5816-8979

### Ethics

Human subjects: Ethical approval was granted by Oxford University's Medical Sciences inter-divisional research ethics committee (MS-IDREC-C2-2015-012) and written informed consent was obtained from all participants prior to the study.

### Decision letter and Author response

Decision letter https://doi.org/10.7554/eLife.37227.012
Author response https://doi.org/10.7554/eLife.37227.013

## Additional files

### Supplementary files

• Transparent reporting form
DOI: https://doi.org/10.7554/eLife.37227.007

### Data availability

The data analysed in this study has been shared as a major dataset.

The following dataset was generated:

| Author(s) | Year | Dataset title | Dataset URL | Database and Identifier |
|---|---|---|---|---|
| Daan B Wesselink, Fiona MZ van den Heiligenberg, Tamar R Makin | 2019 | Effects of arm amputation on motor control | https://osf.io/gmvua/ | Open Science Framework, gmvua |

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
