## [Decision Letter]

Thank you for submitting your article "Obtaining and maintaining cortical hand representation as evidenced from acquired and congenital handlessness" for consideration by *eLife*. Your article has been reviewed by Sabine Kastner as the Senior and Reviewing Editor, and two reviewers.

The reviewers have discussed the reviews with one another and the Reviewing Editor has drafted this decision to help you prepare a revised submission.

Summary:

This study uses fMRI to assess the somatotopic organisation of S1 (and M1) in healthy controls, in amputees and in congenitally one-handed participants. Two approaches are taken. The first is to instruct participants to "move" their (absent) digits. Then a representational similarity analysis is conducted to test for the representational structure across (digit-) conditions. BOLD activity, average representational dissimilarity and "typicality" (a comparison to controls) are similar between controls and amputees, but virtually absent in congenital one-handers. Second, as a control to ensure that this is not simply due to an inability of congenital one-handers in performing the task, a comparison is made using the ipsilateral representation of the other hand, confirming the initial findings.

The experiment builds upon and extends a previous paper by the group published in *eLife*. The difference is that the current paper includes congenital one-handed participants, which allows one to tell whether the representational structure is experience-dependent. The reviewers thought that this is a valuable addition to the previous study. However, there were a number of residual concerns that need to be addressed further.

Essential revisions:

1) For the first finding, the only support is a correlation between one imaging measure (representational typicality) and one questionnaire response (# of independently controllable parts of the hand). Given this, one would expect very large differences between those who can and cannot move their fingers independently – not only in representational typicality, but also in dissimilarity. Surprisingly, the correlation between kinaesthesia and dissimilarity is not reported in the manuscript. As stated in the Introduction, "greater individuation of inputs will induce greater representational dissimilarity." Therefore, if amputees vary on self-reported individuation, then there should be a correlation between these two variables. This needs to be reported. If there isn't a relationship between the two variables, then it makes it quite difficult to interpret the significant correlation between typicality and "kinaesthesia". Second, it is difficult to be confident about correlations with questionnaire data with a small sample size (e.g. the 95% CI for the reported correlation ranges from.33 to.89). Therefore, the relationship between typicality and kinaesthesia would be more convincing if there were other, related correlations that are also significant. (That is, it's hard for us to be confident about one correlation amongst a number of questionnaire and imaging variables, without some additional results that support the same narrative.) Further, the authors stated in the introduction that they would examine "whether persistent representation of a missing hand…will also be observed in amputees with little phantom sensations." Based on this, we were expecting an analysis of the specific individuals with little/no phantom sensations (i.e. some examination of these specific participants, showing that they do (or do not) have a phantom representation similar to intact controls). This should be included in a future submission.

2) The authors show that congenital amelics demonstrate reduced activity and digit discriminability in S1 for phantom movements compared to controls. We share some of the concerns of an earlier reviewer. Given that the authors suggest here and previously (Kikkert et al., 2016) that the S1 activity observed when moving the phantom is due to efference copy, it is unlikely that congenital amelics can generate efference copy for a limb that never existed. However, the lack of efference copy activation in S1 of congenital amelics is not the same as demonstrating that the hand representation has not developed without sensory experience – one of the goals of this paper (see Introduction). That is, there could be a hand representation that exists, but simply hasn't been accessed with the movement task. The manuscript, at times, seems to conflate a lack of activation in S1 for moving a phantom/non-existent hand as evidence for the lack of any hand representation. Although we are sympathetic to this view, we don't think the evidence is there given this single task – and so the claims should be softened.

3) The authors find that controls and amputees did not significantly differ across a number of variables. One example is that typicality did not differ between the two groups. The authors used Bayes factors, using a BF<1/3 as positive evidence in support of the null. Given the editor's concerns regarding exactly what "substantial" evidence means in the context of BFs, one suggestion would be to use frequentist tests designed specifically to examine whether two groups are equivalent (e.g. two one-sided tests for equivalence (TOSTER), Lakens, 2017). This provides a standard p-value that is more straightforward to interpret and is consistent with the (primarily) frequentist tests used throughout the manuscript.

4) The argument for using the ipsilateral hand representation to probe digit structure in the missing hand (Results section) was quite unclear. We understand that Diedrichsen showed a relationship between ipsilateral and contralateral digit representation. But this was in individuals with intact hands. Why would the fact that the "ipsilateral digit representation is a reliable predictor of contralateral hand representation" in two-handers mean that this would also be true for amputees or congenital one-handers? This logic should be laid out.

5) We suggest to briefly explain that the absence of information for the congenital one-handed participants means that the effect is not driven by the visual cue.

6) Is the MDS-analysis cross-validated?

7) To which degree is the typicality analysis influenced by SNR? Does the low dissimilarity of the congenital group (Figure 1) mean that the responses were reproducibly very similar across conditions (i.e. SNR presumably is not responsible for the absence of typicality)?

8) Is this a completely separate dataset than in the original paper? This should be clarified.

[Editors’ note: further revisions were requested before acceptance.]

Thank you for resubmitting your work entitled "Obtaining and maintaining cortical hand representation as evidenced from acquired and congenital handlessness" for further consideration at *eLife*. Your revised article has been favorably evaluated by Eve Marder (Senior Editor) and two reviewers.

The manuscript has been improved but there are some remaining issues that need to be addressed before acceptance, as outlined below: We assume that you will be able to address these issues by editorial revision of the text. Reviewer #1 is now satisfied, and Reviewer 2 agrees that modest textural revisions will be sufficient, and I see no reason why these would need to go back to the reviewers as long as you address the issues.

From Reviewer 2 "With regards to the comment starting with "Further, the authors stated in the Introduction that they would examine.…", my question was regarding whether those with little/no phantom sensations were SIMILAR TO INTACT CONTROLS. However, the author's comments seem to be related to whether these individuals were different from congenital one-handers. First, they say that the intercept for the regression line between phantom vividness and typicality was greater than zero. While this provides evidence for some representational similarity in those with low vividness, it does not mean that they are similar to intact controls. For their second point, they state that the 3 participants with little/no phantom vividness have higher typicality than congenital one-handers. While true, higher typicality than congenital one-handers is not the same as "similar to intact controls". This is not a major point, and doesn't need to be addressed, but could improve the manuscript.

More importantly, with regards to point #4 (and also point #2), we do understand the evidence showing a relationship between ipsilateral and contralateral digit representations in neurologically intact individuals, and the logic that the authors use here. However, no mechanistic argument is presented as to why such a relationship would occur in congenital one-handers, even if they had an intact "missing hand" representations. As it stands, the reader is left with the following implied argument: Since there is a precise correlation between the representational patterns for moving the contralateral hand and ipsilateral hand in the same is, then it must be that representational patterns for the ipsilateral hand assess the representation of the missing hand.

This argument seems to be based on the idea that a strong correlation = causal evidence. Just because this correlation was observed does not mean that moving the ipsilateral hand is necessarily a measure of the contralateral hand representation. For example, one possibility is that in individuals that have had hands at some point in their life, there is a strong relationship between the two representations such that the activity for moving the ipsilateral and contralateral hand in one hemisphere map perfectly. However, there is no mechanistic reason presented to think that this relationship would necessarily hold in those who have never had a hand….it could be that moving the ipsilateral hand does not result in any kind of organized activity in the "missing hand" hemisphere in this population.

Note that I think that this analysis is clever and creative, and I am sympathetic to it. However, as it stands, I don't see the clear mechanistic account for a) why activity for moving the ipsilateral hand is an index of the contralateral hand representation (apart from the correlational argument, which I fear is flawed) and b) why your argument would necessarily hold in congenital one-handers. I believe this could be improved with some mechanistic arguments to support the claim. There is some work on what ipsilateral activation indexes (see Diedrichsen, Wiestler & Krakauer, 2013; prior work on mirror movements)…maybe there is something in this literature that could strengthen the claims?"

---

## [Author Response]

*The experiment builds upon and extends a previous paper by the group published in* eLife*. The difference is that the current paper includes congenital one-handed participants, which allows one to tell whether the representational structure is experience-dependent. The reviewers thought that this is a valuable addition to the previous study. However, there were a number of residual concerns that need to be addressed further.*

We agree with the reviewers’ summary but would like to add that the key innovation extends beyond the study of congenital one-handers. Primarily, whereas all previous studies (both from our own group and beyond) tested people with highly vivid phantom sensations, we have tested a large group of amputees with varying phantom sensations, allowing us to determine whether the experience of having a hand is necessary for the maintenance of the digit maps. Also, our M1 results further complement our original report and are of interest to many people in the field, particularly those working on brain-machine interfaces.

Essential revisions:1) For the first finding, the only support is a correlation between one imaging measure (representational typicality) and one questionnaire response (# of independently controllable parts of the hand). Given this, one would expect very large differences between those who can and cannot move their fingers independently – not only in representational typicality, but also in dissimilarity. Surprisingly, the correlation between kinaesthesia and dissimilarity is not reported in the manuscript. As stated in the Introduction, "greater individuation of inputs will induce greater representational dissimilarity." Therefore, if amputees vary on self-reported individuation, then there should be a correlation between these two variables. This needs to be reported. If there isn't a relationship between the two variables, then it makes it quite difficult to interpret the significant correlation between typicality and "kinaesthesia".

We agree with the reviewers’ reasoning. Since our main prediction concerned typicality and not dissimilarity (based on our original report), we wished to limit the number of comparisons. Yet, based on the reviewers’ suggestion, we have now tested this correlation and added it to the manuscript. As stated below, we found a (trending) positive correlation between kinaesthesia and dissimilarity (rho=.447, p=.063).

Subsection “Missing hand representation in acquired amputees is persistent even after phantom sensations have diminished”: “The correlation between kinaesthesia and dissimilarity in SI approached significance (rho=.447, p=.063).”

Second, it is difficult to be confident about correlations with questionnaire data with a small sample size (e.g. the 95% CI for the reported correlation ranges from.33 to.89). Therefore, the relationship between typicality and kinaesthesia would be more convincing if there were other, related correlations that are also significant. (That is, it's hard for us to be confident about one correlation amongst a number of questionnaire and imaging variables, without some additional results that support the same narrative.)

We agree that, where possible, questionnaires should be validated with other empirical evidence. Since kinaesthesia was one of our two key predictors (along with chronic phantom sensations) we actually asked participants to show us the range of their missing hand movement by mimicking it with their intact hand (see the full study protocol on https://osf.io/gmvua/). This was recorded on video immediately after the scan. Unfortunately, due to a human error, we lost a substantial amount of this data and only have a sub-set of the first 8 acquired amputees tested. These existing videos strongly confirm the self-reports: rates estimated from the videos are strongly correlated with those estimated in the questionnaires (rho=.840). Considering the small sample size, we felt that this extended evidence does not substantially benefit the paper. However, should the editor or reviewers feel differently, we will be happy to include this supporting information.

Regardless, the reviewers make a strong point that we had multiple measures that could potentially be relevant to phantom finger representation. In our first submission, we attempted to deal with this by using a stepwise regression analysis as an exploratory tool. We had to be careful not to overfit the data, so we chose 7 measures that we felt were most appropriate for the missing hand’s typicality measure – time since amputation, age at amputation, kinaesthesia, non-painful phantom vividness (both acute and chronic), and phantom limb pain (both acute and chronic) – as well as typicality of the intact hand’s contralateral activity patterns. The analysis confirmed that kinaesthesia was most relevant to our fMRI measure, allowing us to focus on kinaesthesia in our Results section. We strongly agree that without our regression analysis this focus is not sufficiently justified. Yet, we were also attuned to the original reviewers’ concern that our regression model is not sufficiently reliable for its exploratory purpose.

In the revised manuscript we implemented a bootstrapping approach to provide a measure of confidence for the output of the regression analysis. The final model included an intercept and kinaesthesia as the only regressor (p<.001), with an adjusted R^2^ of.645 (95% confidence interval:.30-.99). Out of 1000 iterations, our measures were included in the final model 96% of the times. We believe that this analysis adds clarity to our Results section without reducing the rigidity of the overall findings. Nevertheless, as in our original finding, we suggest this analysis as an exploratory means only, and do not interpret it further. We have updated the manuscript as follows:

Subsection “Missing hand representation in acquired amputees is persistent even after phantom sensations have diminished”: “Next, we evaluated whether the consistency of hand representation in SI during missing hand movements correlates with amputees’ subjective reports of phantom sensations. We first carried out an exploratory forward stepwise regression with typicality as the dependent variable. The following factors were tested as independent variables: kinaesthesia of phantom sensations (the number of phantom digits perceived as independently moving during the phantom movement task; vividness of nonpainful phantom sensations as experienced both during the study and chronically; intensity of phantom limb pain, as experienced both acutely during the study and chronically; time since amputation; age at amputation and typicality of the intact hand (calculated from the intact hand SI area). The final model (F=19.9, p<.001, adjusted R2=.645 included only kinaesthesia of phantom sensations (β=.07, t=4.46, p<.001) and the intercept (β=.52, t=8.89, p<.001). This regression was submitted to a bootstrapping analysis, allowing us to estimate the consistency of the final model (see Materials and methods). This bootstrapping analysis returned kinaesthesia as the final variable in.963% of the iterations (final model fit: median adjusted R2=.645; 95% CI:.30-.99). The proportion of the other included factors in the final model was: typicality of the intact hand (7.0%); time since amputation (9.5%); age at amputation (9.6%); vividness of nonpainful phantom sensations (acute: 12.2%; chronic: 20.8%); phantom limb pain, acute: 22.7%; chronic: 10.1%).”

Subsection “Statistical Analysis”: “In order to gauge which aspects of phantom sensation and key demographics may relate to the amputees’ representational structure’s typicality, we performed an exploratory forward stepwise regression. The dependent variable was SI typicality in amputees. The following factors were used as independent variables (see also Table 1): typicality of the intact hand (rho; calculated from the intact hand SI area; time since amputation (in years); age at amputation (in years); vividness of nonpainful phantom sensations, as experienced during the study (on a 0-100 scale) and chronically (accounting for both intensity and frequency; Makin et al., 2013b); intensity of phantom limb pain, as experienced acutely during the study, and chronically (as detailed for nonpainful sensations). Only linear factors were considered, i.e. no interaction terms, and the criterion for inclusion was an increase in R^2^ > 0.1. As a large number of predictor variables were included in the model and stepwise regression is generally only recommended for exploratory analysis, we aimed to establish internal replicability using bootstrap resampling (e.g. Thompson, 1995). In particular, we randomly sampled (with replacement) the full data matrix and repeated the stepwise regression 1000 times. We subsequently computed the proportion of bootstrap samples in which each factor which included as well as confidence bounds on the model’s adjusted R^2^. We interpreted high proportion of inclusion (p >.75) as evidence for internal replicability.”

Further, the authors stated in the introduction that they would examine "whether persistent representation of a missing hand…will also be observed in amputees with little phantom sensations." Based on this, we were expecting an analysis of the specific individuals with little/no phantom sensations (i.e. some examination of these specific participants, showing that they do (or do not) have a phantom representation similar to intact controls). This should be included in a future submission.

Indeed, this question was of interest to us. We explored this in our previous submission in two different ways (see Results section of our previous submission). First, we showed that the intercept for the regression line between phantom vividness and typicality was significantly greater than 0 (.75). This means that, based on our population of n=18 amputees, individuals with no phantom sensation are still predicted to show some hand typicality. Secondly, we show that the 3 participants experiencing little to no phantom vividness (below 10/100) showed significantly higher typicality than congenital one-handers. In the revised manuscript, we have reordered the presentation of results and elaborated these results so that they are not easily missed by the readers.

Subsection “Missing hand representation in acquired amputees is persistent even after phantom sensations have diminished”: “Regardless of the positive relationship between kinaesthesia and typicality, amputees with little to no kinaesthetic sensations still showed missing hand representation. As stated above, the regression line between kinaesthesia and typicality included an intercept (rho_intercept_=.52). This was also the case when phantom vividness was the (non-significant) dependent variable (F=.021, p=.89, Adjusted R^2^ =-.061; rho_intercept_ =.75, p<.001). These results predict that even amputees who do not experience any phantom sensations will retain some typical missing hand representation. To test this prediction directly, we examined the 3 amputees in our dataset showing weak to no chronic phantom vividness (below 10/100). Despite not being able to experience clearly their phantom hand when performing the phantom movements task, these individuals showed high typicality (average typicality (rho) =.83). Moreover, when comparing their typicality to that found in the congenital one-handers (who were arguably better matched to this sub-group in terms of task demands), the amputees with diminished phantom sensations showed significantly stronger correlations with the canonical hand structure (Mann-Whitney U = 38, p=.007). Together, these additional analyses confirm that the representational structures’ typicality in SI of amputees is still present in those with little to no phantom or kinaesthetic sensations.”

2) The authors show that congenital amelics demonstrate reduced activity and digit discriminability in S1 for phantom movements compared to controls. We share some of the concerns of an earlier reviewer. Given that the authors suggest here and previously (Kikkert et al., 2016) that the S1 activity observed when moving the phantom is due to efference copy, it is unlikely that congenital amelics can generate efference copy for a limb that never existed. However, the lack of efference copy activation in S of congenital amelics is not the same as demonstrating that the hand representation has not developed without sensory experience – one of the goals of this paper (see Introduction). That is, there could be a hand representation that exists, but simply hasn't been accessed with the movement task. The manuscript, at times, seems to conflate a lack of activation in SI for moving a phantom/non-existent hand as evidence for the lack of any hand representation. Although we are sympathetic to this view, we don't think the evidence is there given this single task – and so the claims should be softened.

We are grateful for this comment, as this is a point that we were very keen to clearly address in our previous submission, and we therefore appreciate the opportunity to improve our delivery. The task constraints imposed on congenital one-handers are similar to those encountered by the few acquired amputees that no longer experience phantom sensations. As highlighted above, these individuals were still able to show higher typicality then congenital one-handers (see above). Regardless, as we highlighted in the original manuscript (and emphasise further in the revised manuscript), we recognise that the task we used might have induced performance differences across the one-handed groups. For this reason, we designed two further analyses that are based on intact hand movements, which all participants could perform equally.

The representational structure of contralateral hand movement can be studied indirectly by examining activity in the same brain region induced by the ipsilateral hand (Diedrichsen et al., 2017). Just to reiterate this point, as an example – left hand movements would evoke strong activity patterns in the contralateral hand area (in the right hemisphere). When participants are moving their ipsilateral (right) hand, that same brain area would show activity patterns, that although smaller, crucially reflect the same representational inter-digit relationships (we elaborate on this key finding below; see comment 4). Because of this, the deprived hand area can be studied based on the ipsilateral intact hand, without imposing unnatural task demands on the congenital one-handers. If congenital one-handers truly have diminished hand representation for their missing hand, their ipsilateral movement should not induce typically structured activity patterns, whereas such activity patterns are expected for the acquired amputees and controls. To test this very specific prediction, we recorded individuated finger movements, which all groups were able to perform equally well. We find that ipsilateral activity following intact hand movement is structured similarly in acquired amputees and controls, but it is reduced in congenital one-handers. In fact, missing hand representation in congenital one-handers was found to be diminished to the point where it is not different from a control visual area.

As such, we are basing our interpretation on multiple converging pieces of evidence: diminished missing hand representation in congenital one-handers using our missing fingers movement task; persistent missing hand representation in acquired amputees with diminished-to-nonexisting phantom sensations; diminished ipsilateral intact hand representation in congneital one-handers; similar ipsilateral digit information in SI and V5 in congenital one-handers, but not amputees/controls.

Nevertheless, we have done our best to frame these results cautiously, e.g. in the abstract and elsewhere we refer to the congenital one-handers missing hand representation as “reduced” or “diminished” rather than non-existent. In the discussion we acknowledge the limitations of our results and call for further research to conclusively resolve this question. We also refrain from inferring that the missing hand representation never developed (as indicated by the reviewers), and instead suggest that the missing hand maps could have originally formed (e.g. based on a genetic blue-print) but later diminished due to insufficient relevant inputs.

In the revised manuscript we have re-written substantial portions of the text (including the abstract), and better structured the Results section (by grouping the findings into sub-sections) to communicate these ideas more clearly. Here are a few examples of revised text addressing this point:

Abstract:

“We used representational similarity analysis in primary somatosensory and motor cortex during missing and intact hand movements. We found that key aspects of acquired amputees’ missing hand representation persisted, despite varying vividness of phantom sensations. In contrast, missing hand representation of congenital one-handers, who do not experience phantom sensations, was significantly reduced. […] We conclude that once cortical organisation is formed, it is remarkably persistent, despite long-term attenuation of peripheral signals.”

Discussion section:

“Still, it is possible that rudimentary missing hand representation, e.g. determined by genetic factors (Miyashita-Lin et al., 1999, Rubenstein et al., 1999), has originally formed in congenital one-handers but later diminished due to lack of consistent sensorimotor input. Our results were insufficiently conclusive to address this important question. Bearing this caveat in mind, our findings suggest that early-life experience is necessary to create typical functional sensory organisation, but not to maintain it.”

3) The authors find that controls and amputees did not significantly differ across a number of variables. One example is that typicality did not differ between the two groups. The authors used Bayes factors, using a BF<1/3 as positive evidence in support of the null. Given the editor's concerns regarding exactly what "substantial" evidence means in the context of BFs, one suggestion would be to use frequentist tests designed specifically to examine whether two groups are equivalent (e.g. two one-sided tests for equivalence (TOSTER), Lakens, 2017). This provides a standard p-value that is more straightforward to interpret and is consistent with the (primarily) frequentist tests used throughout the manuscript.

We appreciate the suggestion to substitute our Bayesian analysis with the frequentist equivalence test TOSTER. After implementing this analysis, we conclude that while the resulting p value might be more intuitive to interpret, the effect size estimate required for conducting the test is conceptually quite tricky. The goal in the TOSTER approach is to specify lower and upper bounds, such that group differences falling within this range are deemed non-meaningful. But how do we choose the a priori equivalence bounds for the existence of a hand representation? One option was using the effect size of the group difference between the controls (who have a hand) and the congenital one-handers (who do not have a hand). Using this criterion, the typicality of the amputees was significantly equivalent to that of the controls (t=3.88, p<.000). But although appropriate for the Bayesian analysis, this boundary might be not suitable for TOSTER. This is because the lower bound should represent the smallest reasonable effect of having a hand, whereas we believe the congenitals do not have hand representation altogether. In other words, the congenital one-handers fall outside this lower boundary. Alternative measures of the lower hand representation boundary can come from the control participants: e.g. contralateral vs. ipsilateral univariate activity (t=6.68; p<.000) or the comparison between SI and V5 typicality (t=6.69; p<0.000). But the fact that these tests produce stronger evidence in favour of the null than our experimental results with the congenital one-handers lead us to doubt this boundary as meaningful as well.

While the TOSTER method clearly supports our main interpretation, in our view, there are two main reason why the Bayesian analysis is better suited to our paper. First, the inclusion of the congenital one-handers in our study was specifically designed to allow us to interpret the persistence of missing hand representation in the acquired amputees. Indeed, the Bayesian analysis allows us to compare the group difference ‘amputees vs controls’ as a measure of having a hand representation. Secondly, the TOSTER method is not quite standard in the field and much less widespread than the Bayesian statistics we have used. Therefore, we prefer implementing the current gold standard for interpreting null results (note that our BF for typicality is.128). Should the editor or reviewers feel that the TOSTER provides substantial further evidence, we will be happy to include it as well.

4) The argument for using the ipsilateral hand representation to probe digit structure in the missing hand (Results section) was quite unclear. We understand that Diedrichsen showed a relationship between ipsilateral and contralateral digit representation. But this was in individuals with intact hands. Why would the fact that the "ipsilateral digit representation is a reliable predictor of contralateral hand representation" in two-handers mean that this would also be true for amputees or congenital one-handers? This logic should be laid out.

We apologise for being unclear. We have adapted the text more explicitly address this crucial point.

Subsection “Diminished missing hand representation in congenital one-handers even when task performance is matched”: “To probe digit structure in the missing hand cortex using an alternative task, we examined ipsilateral hand representation of the intact hand (i.e. in the missing hand cortex). Ipsilateral digit representation in two-handed controls was previously shown to precisely predict contralateral hand representation (Diedrichsen et al., 2018). In other words, individuated digit movements with the ipsilateral hand in sensorimotor cortex produce an identical representational pattern to that of the contralateral hand (though this pattern can be masked by noise). As such, ipsilateral representation of the intact hand provides an indirect assay into the representation of the missing hand, while controlling for task demands across groups. Importantly, all three groups were able to perform the individuated digit movement task, with similar success (see Materials and methods). We compared the intact/dominant inter-digit representational structure in the missing/nondominant hand area of one-handers/controls (respectively). We predicted that persistent missing hand representation in amputees should result in similar ipsilateral representation in their missing hand cortex as controls. If missing hand representation is diminished in congenital one-handers, then ipsilateral representation of their intact hand (in the missing hand area) should show reduced representational features as those found in amputees.”

Discussion section:

“We further show that ipsilateral digit-representation in the missing hand cortex of the intact hand supports the existence of persistent missing hand representation in amputees and diminished representation in congenital one-handers. Previous studies demonstrate that contralateral and ipsilateral hand movements produce identical representational patterns (Diedrichsen et al., 2018). As such, the ipsilateral digit representational pattern is a reliable predictor of contralateral hand representation. This discovery provides us with a unique and novel opportunity to interrogate the information content underlying the “deprived” hand cortex despite the physical absence of the hand. This approach provided converging evidence, but using different task demands, for similar (missing) hand representation for amputees and controls, but not for congenital one-handers.”

5) We suggest to briefly explain that the absence of information for the congenital one-handed participants means that the effect is not driven by the visual cue.

That is a good point. We have added it to our revised manuscript.

Discussion section:

“Using the same task, we were unable to identify similar digit representation for the missing hand of congenital one-handers, demonstrated by significantly reduced pattern typicality compared to both amputees (and even in comparison to those few amputees with little phantom sensations), as well as controls. This result confirms that the persistent hand representation observed in amputees does not reflect mere cognitive task demands (e.g. visual feedback, Kuehn et al., 2018; attention, Puckett et al., 2017).”

6) Is the MDS-analysis cross-validated?

We believe that MDS does not easily lend itself to quantitative analysis, therefore in the manuscript we have relied on cross-validated Mahalanobis distances for all analyses and only used the MDS for visualisation purposes. This is now clarified in the Introduction, in both the Materials and methods section and the Figure legend.

Subsection “fMRI analysis”:

Note that MDS is presented for intuitive visualisation purposes only, and was not used for statistical analysis.

Figure 1:

E) Two-dimensional projection of the representational structure (D) (using multi-dimensional scaling; note that this is included for visualisation purposes only and was not used for statistical analysis).

7) To which degree is the typicality analysis influenced by SNR? Does the low dissimilarity of the congenital group (Figure 1) mean that the responses were reproducibly very similar across conditions (i.e. SNR presumably is not responsible for the absence of typicality)?

This is a good point that we didn’t address so far. When the SNR is low, dissimilarity will tend to 0. Typicality will therefore also be indirectly affected by SNR. In our view, this is not a design flaw, but, in the case of the congenital group, an indication that although the missing hand area of congenital one-handers was activated, it was not activated consistently across scans. To directly address the reviewers’ suggestion, we tested whether the representational patterns evoked by the congenital one-handers are reproducible, by calculating split-half consistency. For each participant, we calculated an RDM for the missing/non-dominant hand using the two odd runs, and one using the two even runs. The correlation between the odd and even RDMs was significantly lower in the congenital one-handers (rho=-.02) compared to both amputees (rho=.41; p_1H-AMP_=.001) and controls (rho=.52; p_1H-CTR_=.001). We note that by splitting the data we are reducing the effectiveness of our analysis (which was designed to rely on 4 scans for cross validation). Nevertheless, the relative difference in split-half consistency between the congenital one-handers and the other two groups indicates that there is no strongly consistent digit information in the missing hand area of the congenital one-handers during our task. We have added the following text to the manuscript to reflect this point:

Subsection “Phantom hand movements elicit typical hand representation in the missing hand area of acquired amputees”: “Although the inter-digit representational structure of congenital one-handers is atypical with respect to canonical hand representation, it is possible that it is still consistent within participants. To explore this idea, we split individual participants’ data to odd and even scans. For each participant, we calculated an RDM in the missing/nondominant hand area using the odd and even runs, and correlated the two RDMs. The correlation between odd and even RDMs was significantly lower in congenital one-handers (rho=-.02) compared to both amputees (rho=.41; p1_H-AMP_=.001) and controls (rho=.52; p_1H-CTR_=.001). We note that by splitting the data we are reducing the effectiveness of our analysis. Nevertheless, the relative reduction in split-half consistency indicates that there is no strongly consistent digit information in the missing hand area of congenital one-handers during this task.”

8) Is this a completely separate dataset than in the original paper? This should be clarified.

We apologise that we did not mention this in our original submission. All three participants from the original paper were scanned again in the current study, namely participants A4, A7, and A2 (respectively to their order in the original paper). After excluding these participants, our two main results still hold: typicality is not significantly reduced in the amputees compared to controls (t=-1.19, df=25, p=.247), and is significantly correlated with kinaesthesia (rho=.70, p=.004). We have added this information to the text.

Subsection “Participants”:

“3 participants in the amputees group tested here also took part in our previous study (Kikkert et al., 2016).”

[Editors’ note: further revisions were requested before acceptance.]

From reviewer 2 "With regards to the comment starting with "Further, the authors stated in the Introduction that they would examine…", my question was regarding whether those with little/no phantom sensations were SIMILAR TO INTACT CONTROLS. However, the author's comments seem to be related to whether these individuals were different from congenital one-handers. First, they say that the intercept for the regression line between phantom vividness and typicality was greater than zero. While this provides evidence for some representational similarity in those with low vividness, it does not mean that they are similar to intact controls. For their second point, they state that the 3 participants with little/no phantom vividness have higher typicality than congenital one-handers. While true, higher typicality than congenital one-handers is not the same as "similar to intact controls". This is not a major point, and doesn't need to be addressed, but could improve the manuscript.

We agree that adding the direct comparison between controls and amputees with little no phantom vividness improves the manuscript and apologise for not having included this initially. The difference between these groups is strongly non-significant (as supported by a Bayesian analysis). We’ve appended the following sentence to the manuscript:

Results section:

“These results predict that even amputees who do not experience any phantom sensations will retain some typical missing hand representation. To test this prediction directly, we examined the 3 amputees in our dataset showing weak to no chronic phantom vividness (below 10/100). Despite not being able to experience clearly their phantom hand when performing the phantom movements task, these individuals showed high typicality (average typicality (rho) =.83). Moreover, when comparing their typicality to that found in the congenital one-handers (who were arguably better matched to this sub-group in terms of task demands), the amputees with diminished phantom sensations showed significantly stronger correlations with the canonical hand structure (Mann-Whitney U = 38, p=.007). Typicality was not different between these 3 amputees and controls (Mann-Whitney U = 29, p=.52, BF=.089). Together, these additional analyses confirm that the representational structures’ typicality in SI of amputees is still present in those with little to no phantom or kinaesthetic sensations.”

More importantly, with regards to point #4 (and also point #2), we do understand the evidence showing a relationship between ipsilateral and contralateral digit representations in neurologically intact individuals, and the logic that the authors use here. However, no mechanistic argument is presented as to why such a relationship would occur in congenital one-handers, even if they had an intact "missing hand" representations. As it stands, the reader is left with the following implied argument: Since there is a precise correlation between the representational patterns for moving the contralateral hand and ipsilateral hand in the same is, then it must be that representational patterns for the ipsilateral hand assess the representation of the missing hand.This argument seems to be based on the idea that a strong correlation = causal evidence. Just because this correlation was observed does not mean that moving the ipsilateral hand is necessarily a measure of the contralateral hand representation. For example, one possibility is that in individuals that have had hands at some point in their life, there is a strong relationship between the two representations such that the activity for moving the ipsilateral and contralateral hand in one hemisphere map perfectly. However, there is no mechanistic reason presented to think that this relationship would necessarily hold in those who have never had a hand….it could be that moving the ipsilateral hand does not result in any kind of organized activity in the "missing hand" hemisphere in this population.Note that I think that this analysis is clever and creative, and I am sympathetic to it. However, as it stands, I don't see the clear mechanistic account for a) why activity for moving the ipsilateral hand is an index of the contralateral hand representation (apart from the correlational argument, which I fear is flawed) and b) why your argument would necessarily hold in congenital one-handers. I believe this could be improved with some mechanistic arguments to support the claim. There is some work on what ipsilateral activation indexes (see Diedrichsen, Wiestler and Krakauer, 2013; prior work on mirror movements)….maybe there is something in this literature that could strengthen the claims?"

We apologise for misunderstanding the concerns of reviewer 2 in our previous round of revisions. We agree that the reviewer raises an important issue that requires further consideration.

Recent evidence showing correspondence between ipsilateral and contralateral representations (across the two hands) sheds some light on the potential mechanism giving rise to ipsilateral hand representation in SI/M1. As we emphasised before, past studies have shown that the activity patterns underlying the ipsi- and contralateral representational structure in one hemisphere are spatially overlapping (Diedrichsen et al., 2013, Diedrichsen et al., 2018). This is generally interpreted as both hands’ representation being facilitated by the same architecture rather than two distinct representations with a correlating structure. In other words, the ipsilateral hand reactivates the patches of cortex associated with the contralateral hand. Strong evidence for this is indeed provided by Diedrichsen et al., (2013, Exp. 2): During bimanual asymmetric movement, SI activity patterns are dominated by contralateral actions only and ipsilateral representation disappears. Most recently, we also showed that ipsilateral representation is only present during active movement (as opposed to passive touch) and not related to uncrossed sensory inflow (Berlot et al., 2018). Taking this evidence together, we have good reason to suggest that in two-handers the motor command to move one hand activates (through the back door) the corresponding representations of the other hand and not a distinct (correlated) representation of the ipsilateral hand. We realise that some of the language used in our manuscript, e.g. emphasising that one representation predicts another, has not reflected this interpretation well and we have made some edits to elaborate on this suggested mechanism.

How does this shared architecture across hands emerge? As pointed out by the reviewers, the two key mechanisms are: Bilateral hand experience, emerging over life, and innate structural connections (e.g. relating to mirroring).

What does this mean for the congenital one-handers? As we see it, there are three possibilities: (1) The deprived cortex develops separate representations for both contralateral (missing) hand and ipsilateral (intact) hand. Here we assume that the overlap and functional interaction seen in controls require coordinated bimanual behaviour, so in one-handers both ipsi- and contralateral hand remain represented distinctly and independently within the deprived cortex. (2) The deprived cortex develops contralateral hand representation (of the missing hand), but intact hand movements do not recruit it, e.g. due to impaired connectivity across hemispheres. (3) The deprived cortex does not develop structured hand representation (neither contralateral nor ipsilateral).

If option 1 were true (though unlikely in light of the research in controls discussed above), then one would expect to find strong ipsilateral representation of the intact hand. Yet, our data does not reflect this hypothesis. Despite typical contralateral dissimilarity for the intact hand (in the intact cortex; not different from controls or acquired amputees’ intact hand; data included in revised manuscript), average ipsilateral dissimilarity (in the deprived cortex) is not higher than our control area, which does not represent hands (Figure 3A).

Option 2 cannot be disproven by our current data, but our previous findings suggest that callosal connections are functionally preserved in congenital one-handers, at least to some extent. We showed that the strength of functional connectivity across the two sensorimotor hand areas correlates with the extent to which congenital one-handers engage their stumps (residual arms) in performing “bi-manual” tasks (Hahamy et al., 2015). Moreover, congenital one-handers who strongly rely on stump usage show normal inter-hemispheric connectivity in comparison to two-handed controls. Since the stump has been shown to strongly activate the missing hand cortex in congenital one-handers (Hahamy et al., 2015; 2017, Makin et al., 2013), it appears that given the right input, the two hemispheres can engage. Please also note that in Makin et al. (2013) we explored structural connectivity changes (using DTI) and did not identify any striking callosal differences between congenital one-handers and controls.

This leaves us with option 3 (no hand representation in the deprived cortex). Taken together, it appears that sensorimotor experience is necessary for developing a hand area, including its connectivity to other areas in the sensorimotor network. We have clarified this both in the Results section and Discussion section, as follows:

Results section:

“To probe digit structure in the missing hand cortex using an alternative task, we examined whether we could observe a representation of the ipsilateral (intact) hand in the missing hand cortex. In two-handed controls, finger movements lead to activity in specific cortical patches in ipsilateral M1 and SI, which tightly correspond to the activity patches engaged in the movement of the mirror-symmetric contralateral finger (Diedrichsen et al., 2013). Indeed, this ipsilateral representation fully overlaps with the representation of the contralateral hand (Diedrichsen et al., 2018). Furthermore, ipsilateral representation disappears completely during asymmetric bimanual finger movements, during which activity in M1 and SI is fully determined by the contralateral hand (Diedrichsen et al., 2013). As such, the ipsilateral representation of one hand is likely elicited due to recruitment of the representation of the contralateral hand (Diedrichsen et al., 2018, Berlot et al., 2018). Ipsilateral representation of the intact hand can therefore provide an indirect assay into the representation of the missing hand, while controlling for task demands across groups.”

Results section:

“If missing hand representation is diminished in congenital one-handers, then ipsilateral representation of their intact hand (in the missing hand area) should show reduced representational features compared to those found in amputees [see Discussion for an alternative mechanism, where the deprived cortex develops separate representations for both the contralateral (missing) and ipsilateral (intact) hands].”

Discussion section:

“We also explored whether we could activate the representation of the missing hand indirectly by movements of the fingers of the intact hand. Previous studies in two-handers have demonstrated that contralateral and ipsilateral hand movements produce identical representational patterns (Diedrichsen et al., 2018). Since this ipsilateral representation is completely overwritten by the contralateral hand if the two hands are engaged in dissociated movements (Diedrichsen et al., 2013, Exp. 2), it has been proposed that the ipsilateral hand reactivates the cortical resources associated with the contralateral hand. Based on this evidence, the ipsilateral digit representation can serve as an indirect measure of the contralateral hand representation.”

Discussion section:

“It might be worth considering whether congenital one-hander’s deprived cortex could have developed separate representations for both the contralateral (missing) and ipsilateral (intact) hand, which are uncorrelated due to anatomical or behavioural differences from two-handers. If this were possible, we would predict maintained, or even stronger, ipsilateral representation of the intact hand. Yet, our data does not reflect this hypothesis. Moreover, poor ipsilateral representation in the deprived cortex does not seem to stem from reduced inter-hemispheric connectivity, which appears to be functionally and structurally preserved in congenital one-handers, depending on lateralisation strategies in daily behaviour (Hahamy et al., 2015, Hahamy et al., 2017, Makin et al., 2013a).”